# The human leukemia virus HTLV-1 alters the structure and transcription of host chromatin in cis

Anat Melamed[1†], Hiroko Yaguchi[1†*], Michi Miura[1], Aviva Witkover[1], Tomas W Fitzgerald[2], Ewan Birney[2], Charles RM Bangham[1*]

[1]Division of Infectious Diseases, Imperial College London, London, United Kingdom; [2]The European Bioinformatics Institute (EMBL-EBI), Cambridge, United Kingdom

**Abstract** Chromatin looping controls gene expression by regulating promoter-enhancer contacts, the spread of epigenetic modifications, and the segregation of the genome into transcriptionally active and inactive compartments. We studied the impact on the structure and expression of host chromatin by the human retrovirus HTLV-1. We show that HTLV-1 disrupts host chromatin structure by forming loops between the provirus and the host genome; certain loops depend on the critical chromatin architectural protein CTCF, which we recently discovered binds to the HTLV-1 provirus. We show that the provirus causes two distinct patterns of abnormal transcription of the host genome in cis: bidirectional transcription in the host genome immediately flanking the provirus, and clone-specific transcription in cis at non-contiguous loci up to >300 kb from the integration site. We conclude that HTLV-1 causes insertional mutagenesis up to the megabase range in the host genome in >$10^4$ persistently-maintained HTLV-1[+] T-cell clones in vivo.
DOI: https://doi.org/10.7554/eLife.36245.001

*For correspondence:
h.yaguchi@imperial.ac.uk (HY);
c.bangham@imperial.ac.uk (CRMB)

†These authors contributed equally to this work

**Competing interests:** The authors declare that no competing interests exist.

## Introduction

The dynamics and higher-order folding of chromatin play a critical role in gene regulation. Higher-order chromatin structure *is determined by* several factors, among which *the best characterized* is CCCTC-binding factor (CTCF) (*Ong and Corces, 2014*). CTCF binds a non-palindromic 20-nucleotide DNA motif at ~50,000 sites in the human genome, and acts chiefly (*Phillips and Corces, 2009*) by regulating the formation of chromatin loops of ~100 kb to ~2 Mb, which control the contacts made between promoters and enhancers and so regulate gene expression (*Bartman et al., 2016; Gibcus and Dekker, 2013; Schwarzer and Spitz, 2014*). Aberrant higher-order chromatin organization can result in abnormal patterns of transcription, and mutations in CTCF are linked to human disease (*Corces and Corces, 2016; Flavahan et al., 2016; Hnisz et al., 2016*).

Recently we found (*Satou et al., 2016*) that CTCF binds to a nucleotide motif in the Human T lymphotropic virus type 1 (HTLV-1; also known as the human T-cell leukemia virus), when HTLV-1 is integrated - as the provirus - into the host cell genome. HTLV-1 is a primate retrovirus that infects ~10 million people in the tropics and subtropics (*Bangham, 2018*). The infection is asymptomatic in ~90% of human hosts; the remaining 10% of HTLV-1-infected hosts develop either a chronic inflammatory disease, most commonly involving the central nervous system, or an aggressive malignancy of CD4[+] T cells known as adult T-cell leukemia/lymphoma (ATL). Unlike HIV-1 infection, there remains no satisfactory treatment for either the inflammatory or malignant diseases caused by HTLV-1.

A typical host of HTLV-1 carries between $10^4$ and $10^5$ clones of infected T cells, each clone carrying a single copy of the provirus in a unique genomic location (*Laydon et al., 2014*). The large number of HTLV-1-infected clones appears to be established early in infection, after which persistent

clonal proliferation maintains a stable hierarchy of HTLV-1-infected clones for the remainder of the host's life. The viral regulatory proteins, Tax and HBZ (HTLV-1 bZIP), which are encoded respectively by the sense and antisense strands of the provirus, play indispensable roles in pathogenesis (*Matsuoka and Jeang, 2011a*). Tax is a transcriptional transactivator which dysregulates many host genes, while HBZ acts as a negative regulator of Tax-mediated host gene transcription and viral expression. Both Tax and HBZ contribute to persistent proliferation of the infected T cell in vivo, and it is now thought that the consequent accumulation of replicative mutations is a key driver in HTLV-1 oncogenesis. However, until recently (*Cook et al., 2014*; *Kataoka et al., 2015*; *Rosewick et al., 2017*) insertional mutagenesis has not been considered important in causing ATL.

The observation that CTCF binds to the HTLV-1 provirus (*Satou et al., 2016*) raised the hypothesis that CTCF bound to the provirus can form abnormal chromatin loops by dimerizing with CTCF in the flanking host genome. Using chromosome conformation capture (3C), we previously demonstrated the presence of a single CTCF-dependent chromatin loop in a long-term in vitro T cell line (*Satou et al., 2016*). In the present study we extended this analysis, using circular chromosome conformation capture (4C) and RNA-seq, to examine systematically the impact of the HTLV-1 provirus on the structure and expression of the host genome in T cell clones isolated from HTLV-1-infected subjects. We show that the HTLV-1 provirus forms reproducible abnormal chromatin contacts with sites in the host genome in cis as far as 1.4 Mb from the provirus. Some of these abnormal chromatin contacts depend on CTCF binding to the provirus. Further, we demonstrate clone-specific deregulation of host transcription in cis both immediately flanking the integrated provirus (up to 50 kb upstream and downstream) and at distant sites as far as 300 kb from the provirus. Since HTLV-1 is integrated in $>10^4$ different genetic locations in a typical host, and there are tens of thousands of CTCF-binding sites (CTCF-BS) in the human genome (*Krijger and de Laat, 2016*), these results imply that HTLV-1 has the potential to cause deregulation of host transcription at a very large number of loci in each infected host.

## Results

In this paper, we refer to loci in the host genome relative to the orientation of the HTLV-1 provirus. Thus, a locus 'downstream' of the provirus is located 3′ to the 3′ LTR, whether the provirus is integrated in the positive or negative sense in the host genome.

### HTLV-1 forms chromatin loops with the flanking host genome

To test the hypothesis that HTLV-1 integration alters the host chromatin structure, we performed a genome-wide search for chromosomal positions that contact the HTLV-1 provirus, using a modified version of circular chromosome conformation capture (4C). 4C is a powerful tool to study the 3D chromatin looping between a specified genomic region (the 'viewpoint') with respect to the rest of the genome. The standard 4C protocol has limitations in that it is only semiquantitative (*Denker and de Laat, 2016*). To improve the quantification, we adapted our protocol for quantification of HTLV-1 integration sites (*Gillet et al., 2011*; *Gillet et al., 2017*). We followed the 4C protocol described by van de Werken et al. (*van de Werken et al., 2012*), but instead of using a second restriction enzyme, we sonicated the library, added adapters and performed ligation-mediated PCR (see Materials and methods). This modification confers two advantages. First, it precludes the bias towards detection of chromatin contacts that lie near a given restriction site. Second, the amplicon length serves as a unique molecular identifier, enabling relative quantification (*Gillet et al., 2017*) of the chromatin contacts. We refer to this modified method as quantitative 4C (q4C); a similar approach (UMI-4C) has been described by others (*Schwartzman et al., 2016*).

We applied q4C to test the hypothesis that the HTLV-1 provirus forms chromatin contacts with the host genome in a series of T cell clones isolated by limiting dilution from circulating CD4$^+$ T lymphocytes of HTLV-1-infected individuals (*Cook et al., 2012*) (*Table 1*; *Supplementary file 1*). Each clone has a single copy of the provirus in a unique integration site. As the viewpoint in q4C, we used the 679 bp NlaIII fragment containing the proviral CTCF-BS (*Figure 1A*). Chromatin contacts were identified using a protocol based on a hidden Markov model (Materials and methods).

We detected reproducible q4C peaks (long-range chromatin contacts between the provirus and the host genome) in 9 of the 10 infected T-cell clones examined (*Figure 1B,C*; *Figure 1—figure supplement 1–10*). The number of peaks per clone varied between 0 and 15, with a median of 3 peaks

**Table 1.** T cell clones used.

| Subject | Clone(s) |
| --- | --- |
| TBJ | 3.60, 3.83 |
| TCX | 8.13, 8.8 |
| TCT | 10.1 |
| TBW | 11.50, 11.63, 11.65, 13.50(U) |
| TBX | TBX4B |
| HAY | 6.25, 6.30(U) |

DOI: https://doi.org/10.7554/eLife.36245.014

per clone (*Figure 1C*); There were significantly more peaks downstream of the integration site than upstream (p=0.03, Chi-squared goodness-of-fit test; *Figure 1—figure supplement 11A*). The distance between identified peaks and the provirus of the respective clone varied between 12.9 kb and 1.4 Mb, with a median of 85 kb (*Figure 1D*). The distance between each peak and the integration site did not significantly differ between upstream and downstream peaks (p=0.13, Wilcoxon test; *Figure 1—figure supplement 11B*).

We wished to identify whether the HTLV-1 provirus makes preferential contacts with the host genome in cis. The provirus is present in a single copy per cell (*Figure 2A*) (*Cook et al., 2012*). First, to determine whether the q4C reads were derived from a single chromosome (i.e. were monoallelic) or from both homologous chromosomes (biallelic), we identified single-nucleotide polymorphisms (SNPs) present in the respective donor subject by whole-genome sequencing (see Materials and methods). The alleles identified at heterozygous SNPs in q4C reads demonstrated that the observed ligation events were confined to a single strand (i.e. were monoallelic), with a range of at least 4 Mb from the provirus (*Figure 2B*).

Next, we used the heterozygous SNPs to distinguish the chromosome carrying the provirus from its homologous chromosome, using computationally-determined phased haplotypes (see Materials and methods). In this way, the infected chromosome could be distinguished from the uninfected chromosome for at least 100 kb either side of the integration site in 8 of the 10 clones (*Figure 2C*; *Figure 2—figure supplement 1–5*). To validate the haplotype-calling, we identified heterozygous SNPs in long-range PCR products, amplified either between the provirus and the host genome (infected haplotype) or across the proviral integration site (uninfected haplotype) (*Figure 2C*; *Figure 2—figure supplement 1–5*). The results (*Figure 2C*) showed that the reproducible contacts between the host genome and the HTLV-1 provirus were exclusively made in cis, that is, on the infected chromosome.

## Certain long-range chromatin contacts are CTCF-dependent

We wished to test the hypothesis that the observed abnormal long-range chromatin contacts are associated with the presence of CTCF-BS identified in the T cell clones by ChIP-seq. The results showed that that CTCF-BS were enriched at the chromatin contacts in the host genome: ~50% of q4C peaks overlapped with at least one CTCF-BS (*Figure 3A*); in 10% of peaks there were two CTCF-BS. The frequency of CTCF-BS overlapping the observed 4C peaks was significantly higher than random expectation (p=4.73 * $10^{-4}$, Fisher's exact test; *Figure 3—figure supplement 1*). Consistent with recent findings by others (*de Wit et al., 2015*; *Sanborn et al., 2015*), where a peak overlapped a CTCF-BS the viral and host CTCF binding motifs were present in convergent or tandem orientation in 80% of cases (*Figure 3B*). However, not all nearby tandem or convergent host CTCF-BS formed detectable contacts with the provirus: see, for example, clones 8.13 and TBX4B (*Figure 1—figure supplement 5* and *8*, respectively). The presence of a CTCF-BS was associated with a significantly greater observed q4C peak height (p=0.025, Wilcoxon test), and there was a significant positive trend between the q4C peak height and the number of CTCF sites within the peak (*Figure 3C*; p=0.016, Spearman's test). Finally, there was a significant positive correlation between the number of observed chromatin contacts in a clone and the number of CTCF-BS within 0.5 Mb of the integration site (*Figure 3D*; Pearson's correlation test); this correlation remained significant up to 1.16 Mb from the provirus.

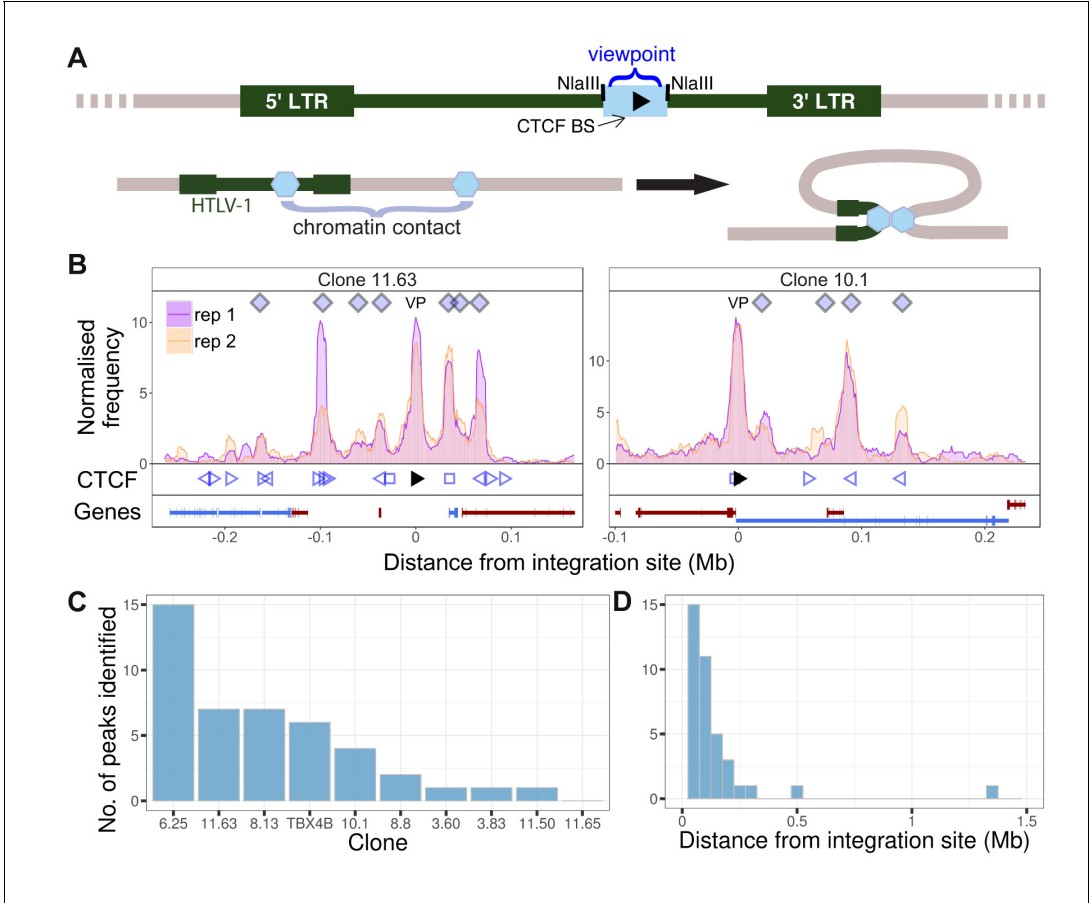

**Figure 1.** HTLV-1 forms distant contacts with the host genome. (**A**) Upper line: the HTLV-1 genome (green), with a long terminal repeat (LTR) at each end, is integrated into a clone-specific site in the human genome (grey). The q4C viewpoint (blue rectangle) is the NlaIII fragment within the HTLV-1 genome (nucleotide residues 6564–7246) which contains the CTCF binding site (CTCF-BS; black arrowhead). Lower line: the CTCF-BS (blue hexagon) in the provirus can dimerize with a CTCF-BS in the flanking host genome. (**B**) Chromatin contacts identified by q4C in two different clones. For each clone, the top panel depicts the q4C profile in the 5′ and 3′ host genome flanking the provirus (two biological duplicates), quantified as the normalized frequency of ligation events in overlapping windows (window width 10 kb, step 1 kb). On the horizontal axis, positive values denote positions downstream of the provirus (i.e. lying 3′ of the 3′ LTR); negative values denote upstream position. VP – viewpoint in q4C (proviral integration site). Diamonds mark the positions of reproducible chromatin contact sites called by the peak caller (Materials and methods). CTCF panel – open arrowheads denote positions of CTCF-BS; the filled arrowhead denotes the CTCF-BS in the provirus. Genes panel shows RefSeq protein-coding genes in the flanking host genome. The q4C profiles of remaining clones are shown in *Figure 1—figure supplement 1–10*. (**C**) Number of detected peaks in each clone. (**D**) Distance from detected q4C peaks to the respective proviral integration site.

DOI: https://doi.org/10.7554/eLife.36245.002

The following figure supplements are available for figure 1:

**Figure supplement 1.** q4C and RNASeq data aligned – clone 6.25.

DOI: https://doi.org/10.7554/eLife.36245.003

**Figure supplement 2.** q4C and RNASeq data aligned – clone 10.1.

DOI: https://doi.org/10.7554/eLife.36245.004

**Figure supplement 3.** q4C and RNASeq data aligned – clone 8.8.

DOI: https://doi.org/10.7554/eLife.36245.005

**Figure supplement 4.** q4C and RNASeq data aligned – clone 3.83.

DOI: https://doi.org/10.7554/eLife.36245.006

**Figure supplement 5.** q4C and RNASeq data aligned – clone 8.13.

DOI: https://doi.org/10.7554/eLife.36245.007

**Figure supplement 6.** q4C and RNASeq data aligned – clone 11.50.

DOI: https://doi.org/10.7554/eLife.36245.008

**Figure supplement 7.** q4C and RNASeq data aligned – clone 11.63.

DOI: https://doi.org/10.7554/eLife.36245.009

*Figure 1 continued on next page*

*Figure 1 continued*

**Figure supplement 8.** q4C and RNASeq data aligned – clone TBX4B.
DOI: https://doi.org/10.7554/eLife.36245.010
**Figure supplement 9.** q4C and RNASeq data aligned – clone 11.65.
DOI: https://doi.org/10.7554/eLife.36245.011
**Figure supplement 10.** q4C and RNASeq data aligned – clone 3.60.
DOI: https://doi.org/10.7554/eLife.36245.012
**Figure supplement 11.** Position of q4C peaks relative to the HTLV-1 provirus. We defined 'upstream' peaks as q4C peaks that lie on the 5' side of the 5' LTR of the HTLV-1 provirus, and 'downstream' peaks as those which lie 3' to the 3' LTR.
DOI: https://doi.org/10.7554/eLife.36245.013

We then tested whether CTCF binding to the provirus is required for formation of provirus-host contacts. We used a CRISPR-modified cell line (ED) (*Satou et al., 2016*), derived from an individual with the HTLV-1-associated malignancy adult T-cell leukemia (ATL) (*Figure 3E*). We also applied CRISPR-Cas9 ribonucleoprotein transfection (*Schumann et al., 2015*) to knock out the CTCF-BS in a non-malignant T-cell clone (TBX4B; *Figure 3F*). We carried out q4C analysis on cells from the wild-type (WT) clone, and the mutant (Mut) clone containing a mutated CTCF-BS. The results show that the loss of CTCF binding was associated with a loss of 5 of the eight observed contacts; three of the five lost contacts overlapped a CTCF-BS in the host.

## HTLV-1 alters contiguous host transcription

The results obtained by q4C demonstrated that HTLV-1 integration can alter host chromatin looping. We wished to investigate the impact of the HTLV-1 provirus on transcription in the host genome both immediately flanking the provirus and at distant loci. We carried out strand-specific mRNA-sequencing (RNA-seq) on the HTLV-1-infected T cell clones, and quantified the density of reads mapping to discrete 1 kb windows up to 2 Mb from the respective proviral integration site. The results showed upregulated transcription in the host genome immediately flanking the HTLV-1 provirus, in all clones examined (*Figure 4A,B*; *Figure 1—figure supplement 1–10*, *Figure 4—figure supplement 1*). This upregulated transcription was observed either upstream or downstream of the provirus, or both, with a predominant increase downstream in the same sense as the HTLV-1 plus-strand and upstream in the opposite sense (*Figure 4C*). We then performed this analysis separately to compare the clones with high HTLV-1 plus-strand expression and those with low plus-strand expression (defined respectively as those clones with a *tax* read intensity in the RNA-seq above or below the median intensity of all clones). The results showed that, whereas abnormal upstream anti-sense expression was present in most clones, an increase in same-sense transcription was specific to those clones with high plus-strand expression (*Figure 4D*).

The observed upregulation of transcription was frequently intergenic, but we also observed instances of clone-specific gene expression. For example, in clone TBX4B, HTLV-1 is inserted between exons of the gene *PNPLA* (*Figure 3F*). This gene was not expressed in the other T cell clones, but was highly expressed in TBX4B both downstream and upstream of the integration site, in the same sense as the proviral plus-strand. The presence of abnormal same-sense host transcription upstream of the 5' LTR suggests that the transcription was driven by the proviral enhancer.

## Transcription is altered at non-contiguous sites

In addition to the abnormal transcription immediately flanking the provirus, there were frequent examples of clone-specific transcription in regions of the host genome not contiguous with the provirus (*Figure 4B*. *Figure 4—figure supplement 1*). For example, in clone 8.8, there was upregulation of host transcription both flanking the integration site and in the gene *TBC1D4* (a Rab GTPase-activating protein) ~44 kb upstream; no transcripts were detected in the intervening host genome (*Figure 4B*, left). Non-contiguous transcription also occurred at the downstream contact site made with the provirus, resulting in aberrant splicing of the gene *UCHL3* to produce putative novel *UCHL3* transcripts (*Figure 4—figure supplement 2B*).

In clone 3.60, abnormal non-contiguous transcription was found downstream of the provirus (*Figure 4B*, right) between the provirus and the contact (indicated by the q4C peak) formed with the host genome. This region contains a gene (*SULT1B1*) in the negative strand of the genome; the

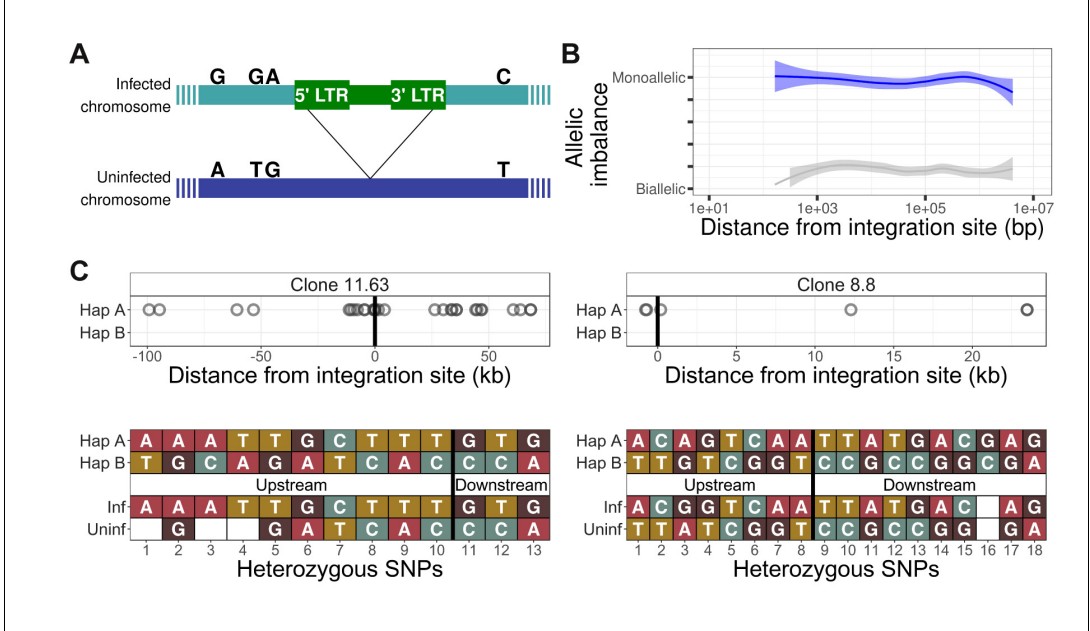

**Figure 2.** The HTLV-1 provirus makes chromatin contacts in cis with the infected chromosome. (**A**) The HTLV-1 provirus is present in one copy per cell. The infected chromosome (green) can be distinguished from the uninfected homologous chromosome (dark blue) by heterozygous single-nucleotide polymorphisms (SNPs), marked by the nucleotides above each chromosome. (**B**) The frequency of allele usage in unique q4C reads containing heterozygous SNPs (at least two reads per position) was measured, to quantify the degree of allelic imbalance, that is the degree of monoallelic usage present in q4C reads at a heterozygous SNP. Allelic imbalance ranges between 0 (biallelic, i.e. half of reads come from each allele) and 0.5 (monoallelic, i.e. all reads from one allele only). The dark blue line (above) shows the range of allele usage in the q4C reads; the light grey line (below) shows the allele usage for the same SNPs in the whole-genome sequencing reads. Curves were computed using LOESS regression. (**C**) The infected chromosome was distinguished from the homologous uninfected chromosome using q4C data (top panel) and chromosome-specific PCR (bottom panel). Top panel - heterozygous SNPs in DNA were phased computationally to identify the two haplotypes (A and B) (Materials and methods); the alleles present in q4C data were then assigned to the respective haplotype (circles). On the horizontal axis, positive values denote positions downstream of the provirus, and negative values denote positions upstream. Within at least 100 kb, all identified heterozygous SNP alleles mapped to only one of the two haplotypes. Bottom panel – haplotype assignment was confirmed using haplotype-specific PCR. Each nucleotide shown is a heterozygous SNP within 5 kb of the proviral integration site, identified in the respective clone by whole-genome sequencing. The SNPs were then mapped to the respective haplotype by Sanger sequencing of long-range products amplified by PCR either between the provirus and host genome (inf – infected haplotype) or across the provirus (uninf – uninfected haplotype). Further examples are shown in *Figure 2—figure supplement 1–5*.

DOI: https://doi.org/10.7554/eLife.36245.015

The following figure supplements are available for figure 2:

**Figure supplement 1.** Identification of infected chromosomes - clone 11.50
DOI: https://doi.org/10.7554/eLife.36245.016
**Figure supplement 2.** Identification of infected chromosomes - clone 11.65.
DOI: https://doi.org/10.7554/eLife.36245.017
**Figure supplement 3.** Identification of infected chromosomes - clone 6.25.
DOI: https://doi.org/10.7554/eLife.36245.018
**Figure supplement 4.** Identification of infected chromosomes - clone 8.13.
DOI: https://doi.org/10.7554/eLife.36245.019
**Figure supplement 5.** Identification of infected chromosomes - clone TBX4B.
DOI: https://doi.org/10.7554/eLife.36245.020

abnormal transcription observed in clone 3.60 was present in the positive strand with alternative splicing, resulting in a putative novel transcript (*Figure 4—figure supplement 2A*). These observations demonstrate that the provirus can alter transcription in cis both within and between genes, and alter splicing, producing novel transcripts.

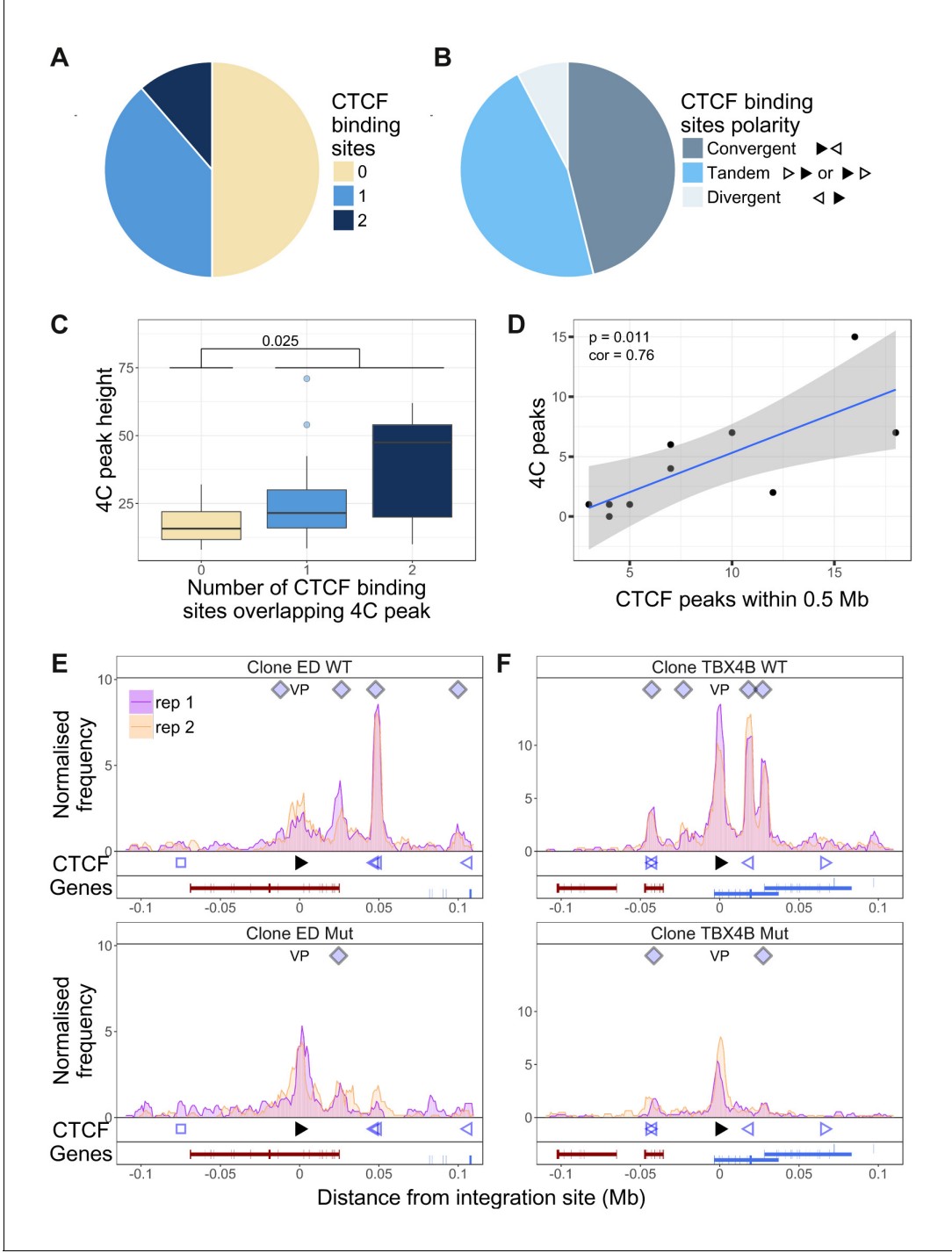

**Figure 3.** Dependency of virus-host contacts on CTCF binding. (**A**) Of 44 contacts identified by q4C in the clones examined, 22 contained one CTCF-BS (N = 17) or two CTCF-BS (N = 5); the remaining 22 contacts did not contain a CTCF-BS. The presence of one or more CTCF-BS in observed q4C peaks was higher than expected by chance (p=4.73 * $10^{-4}$, Fisher's exact test; see also *Figure 3—figure supplement 1*). (**B**) The polarity was determined of the proviral CTCF-BS (filled arrowhead) and the host CTCF-BS (open arrowhead). Of the CTCF-containing peaks whose polarity could be determined, convergent orientation (possible only for downstream peaks) was found in 46% of peaks, divergent orientation (possible only for upstream peaks) in 8% of peaks, and tandem orientation (possible for either upstream or downstream peaks) in 46% of peaks. (**C**) Distribution of q4C peak height (mean number of ligation events between replicates identified in each peak) in peaks containing 0, 1 or 2 CTCF-BS (coloured as in panel A): peaks that contained at least one CTCF-BS were significantly higher than those that lacked a CTCF-BS (p=0.025, Wilcoxon test). In addition, there was a significant correlation between mean q4C peak height and the number of overlapping CTCF-BS (p=0.0156, Spearman's rank correlation test) (not illustrated). (**D**) The number of observed contacts was positively correlated with the number of CTCF-BS within 0.5 Mb of the proviral integration site (p=0.011,

*Figure 3 continued on next page*

*Figure 3 continued*

Pearson's correlation test). (**E–F**) q4C analysis was carried out on a clone from an ATL-derived cell line (**E**) and a T-cell clone (**F**), respectively either the wild-type cells (WT; top panels) or after CRISPR-Cas9 knockout of the proviral CTCF-BS (bottom panels). The vertical axis shows the normalized number of q4C ligation events (overlapping windows 5 kb wide, with 1 kb steps). The bottom track in each panel shows the position of known RefSeq protein-coding genes; in clone TBX4B the provirus is inserted between exons of the gene *PNPLA* (shown in blue; see Results section).

DOI: https://doi.org/10.7554/eLife.36245.021

The following figure supplement is available for figure 3:

**Figure supplement 1.** CTCF-BS occupancy at 4C peaks is higher than random expectation.

DOI: https://doi.org/10.7554/eLife.36245.022

## HTLV-1 alters host transcription in cis

To test the hypothesis that the observed abnormal transcription of the host genome was confined to the chromosome carrying the provirus, we quantified the allelic imbalance in the SNPs identified in RNA-seq reads up to 2 Mb from the integration sites. The results showed that in heterozygous SNPs identified in the clone-specific transcripts, transcription was predominantly homozygous, that is, monoallelic (*Figure 5A*).

To test whether this monoallelic transcription was derived from the infected chromosome or its homologous chromosome, we used the haplotype-calling approach described above to assign heterozygous SNPs present in the RNA-seq reads within 100 kb of the proviral integration site to either the infected or the uninfected chromosome. The results (*Figure 5B,C,D*) showed that, where the transcripts could be assigned to one chromosome, the observed clone-specific transcripts were overwhelmingly derived from the infected chromosome, whereas transcripts that were not specifically upregulated were expressed from either chromosome at a similar frequency (*Figure 5B*, left side of left panel; *Figure 5C,D*).

## Discussion

The mammalian genome is not randomly arranged in the nucleus, but is folded in a highly ordered manner at a series of successively larger spatial scales (*Bickmore, 2013*). At the smallest scale, chromatin is organized in a series of reproducible loops that are formed by bringing together specific genomic loci which are separated on the linear genome by up to ~2 Mb (*Dekker and Heard, 2015*; *Dixon et al., 2012*; *Nora et al., 2012*). The zinc finger protein CTCF plays a central part in establishing and maintaining these chromatin loops: the non-palindromic CTCF-BS is found at the borders of ~80% of loops. Certain other chromatin-associated proteins, such as PRC1 (*Schoenfelder et al., 2015*), and transcription factors including AP-1 (*Phanstiel et al., 2017*) and YY1 (*Beagan et al., 2017*), can also cause looping of chromatin. The resulting chromatin loops in turn play a critical part in controlling gene expression, by regulating the contacts made between enhancers and promoters (*Bartman et al., 2016*; *Gibcus and Dekker, 2013*; *Schwarzer and Spitz, 2014*). Disruption of chromatin loops can deregulate gene expression and cause developmental abnormalities (*Lupiáñez et al., 2015*) or diseases such as cancer (*Corces and Corces, 2016*; *Flavahan et al., 2016*; *Hnisz et al., 2016*).

The discovery that the HTLV-1 provirus binds CTCF (*Satou et al., 2016*) therefore raised the hypothesis that the provirus forms chromatin loops with the neighbouring host genome and thereby deregulates host transcription. To test this hypothesis, we used a panel of non-malignant CD4$^+$ T cell clones naturally infected with HTLV-1. Each clone carries a single copy HTLV-1 provirus in a known genomic location (*Cook et al., 2012*); all 10 clones studied were competent to express the plus-strand proviral genes except clones 6.25 and 8.13.

The results of chromosome conformation analysis (q4C) presented here reveal reproducible contacts between the HTLV-1 provirus and the flanking host genome at least 1.4 Mb from the integration site. Each clone has a unique pattern of chromatin contacts, depending on the site of integration of the provirus in the host genome. The allelic bias observed in the q4C data, with the preferential detection of SNPs on the chromosome carrying the provirus, indicates that the provirus makes preferential contacts in cis with host chromatin at distances at least 4 Mb from the provirus. Preferential contacts in cis may extend beyond this distance, but detection is likely to be limited by the sensitivity of the q4C technique.

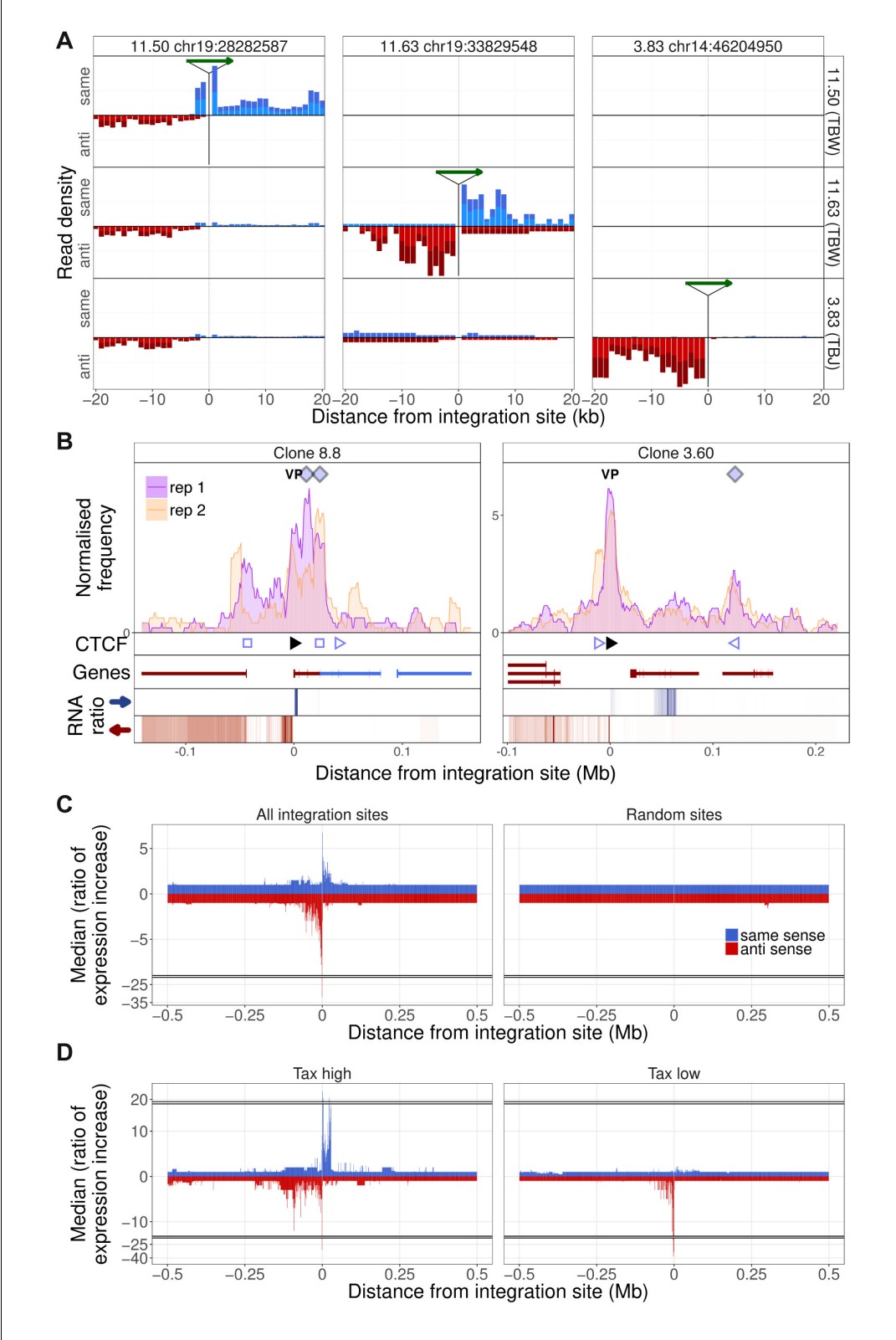

**Figure 4.** Integration site-specific upregulation of host transcription. (**A**) In each column, the green arrow indicates the HTLV-1 proviral integration site in the clone indicated at the top of the column. Each row shows the the transcription density (normalized RNA-seq read count) flanking that genomic position in the clone indicated at the right-hand side. In each case, transcription orientation and positions are shown relative to the integrated provirus. Read density shown in blue shows transcription in the same orientation as the proviral plus-strand; red shows transcription in the antisense orientation

*Figure 4 continued on next page*

*Figure 4 continued*

to the proviral plus-strand. (B) q4C profiles of two clones aligned with the transcription density within 300 kb of the proviral integration site. The RNA ratio shows the ratio of transcription density in a given bin (number of reads in 1 kb bin/total number of reads in sample) in the target clone, divided by the median expression density of all clones in that bin. Colours represent expression in the same sense (blue) or opposite sense (red) to the HTLV-1 plus-strand. Data on the remaining clones are shown in *Figure 1—figure supplement 1–10*. (C) Left panel: median ratio of transcription density of all clones, aligned on the integration site (1 kb bins, up to 0.5 Mb from the integration site). Right panel: median ratio of transcription density at 10 genomic positions, selected at random from a gap-excluded hg19 reference genome. (D) Analysis carried out as in panel C, separately for clones expressing HTLV-1 plus-strand transcripts at a level greater than (left panel) or less than (right panel) the median of all clones.

DOI: https://doi.org/10.7554/eLife.36245.023

The following figure supplements are available for figure 4:

**Figure supplement 1.** Upregulation of transcription within 100 kb of integration site.
DOI: https://doi.org/10.7554/eLife.36245.024

**Figure supplement 2.** Examples of clone-specific aberrant transcription and splicing.
DOI: https://doi.org/10.7554/eLife.36245.025

Four observations indicate that certain chromatin contacts made between the provirus and the host genome depend on the presence of CTCF. First, in each of two clones (ED and TBX4B; *Figure 3E,F*), knocking out the CTCF BS in the provirus without altering the coding sequence of the *tax* gene abrogated a major contact with a site in the host genome, respectively ~50 and 17 kb downstream, where CTCF was shown to bind. Second, of the 44 host loci identified by the peak-calling algorithm as making contact with the provirus, 22 contained one or more CTCF-BS (p=4.73 * $10^{-4}$, Fisher's exact test). Third, consistent with recent observations made by others (*de Wit et al., 2015*; *Sanborn et al., 2015*), 12 of the 13 (92.3%) CTCF-BS in the host contact sites whose orientation could be ascertained were oriented towards the CTCF-BS in HTLV-1. Fourth, both the number and height of q4C peaks were correlated with the number of CTCF-BS in the host genome (*Figure 4C,D*). However, not all host chromatin contact sites contain CTCF-BS, and certain peaks remained detectable by q4C after knock-out of the proviral CTCF-BS. These observations are consistent with the findings that certain chromatin-associated proteins other than CTCF can also give rise to chromatin looping (*Schoenfelder et al., 2015*; *Phanstiel et al., 2017*; *Beagan et al., 2017*).

The formation of chromatin loops between the HTLV-1 provirus and the host genome in turn raised the possibility that host transcription is deregulated. At least two mechanisms of such deregulation can be suggested. First, the abnormal chromatin loops might alter the normal contacts made between enhancers and promoters in the host genome, either creating new contacts or abrogating pre-existing contacts. Second, the new chromatin loop might bring the proviral LTR near a host promoter and cause abnormal transcription from the promoter.

We show here that HTLV-1 can deregulate host transcription both at sites contiguous with the provirus and at non-contiguous, distant sites. Again, each clone has a unique pattern of transcription. In the host genome immediately flanking the provirus, we detected frequent abnormal (clone-specific) transcription, predominantly in the opposite transcriptional sense to the provirus upstream of the 5′ LTR and, to a lesser extent, in the same transcriptional sense downstream of the 3′ LTR. The observed abnormal transcription involved both intergenic regions and identified host genes (*Figure 4B*, *Figure 1—figure supplement 1–10*, *Figure 4—figure supplement 2*); in addition, we observed examples of abnormal transcription and abnormal splicing of host genes (*Figure 4—figure supplement 2*).

Kataoka et al. (*Kataoka et al., 2015*) and Rosewick et al. (*Rosewick et al., 2017*) have reported evidence of transcription initiated on the minus strand of the provirus and continuing upstream into the host genome in transformed lymphocyte clones carrying HTLV-1 or the related bovine leukemia virus (BLV). The present results are consistent with these recent reports, demonstrating that abnormal host transcription flanking the provirus in cis is a general feature of HTLV-1 infection. Our results further suggest that an important mechanism of the observed abnormal transcription is chromatin looping between the provirus and the host genome. The abnormal transcription is bidirectional: both downstream of the provirus in the same sense as the proviral plus-strand, and upstream in the opposite sense. The observation of clone-specific transcription upstream of the 5′ LTR in the plus-strand sense (*Figure 4A*), especially in high *tax*-expressing cells (*Figure 4D*), suggest that the HTLV-1 promoter/enhancer can drive abnormal expression from nearby transcription start sites.

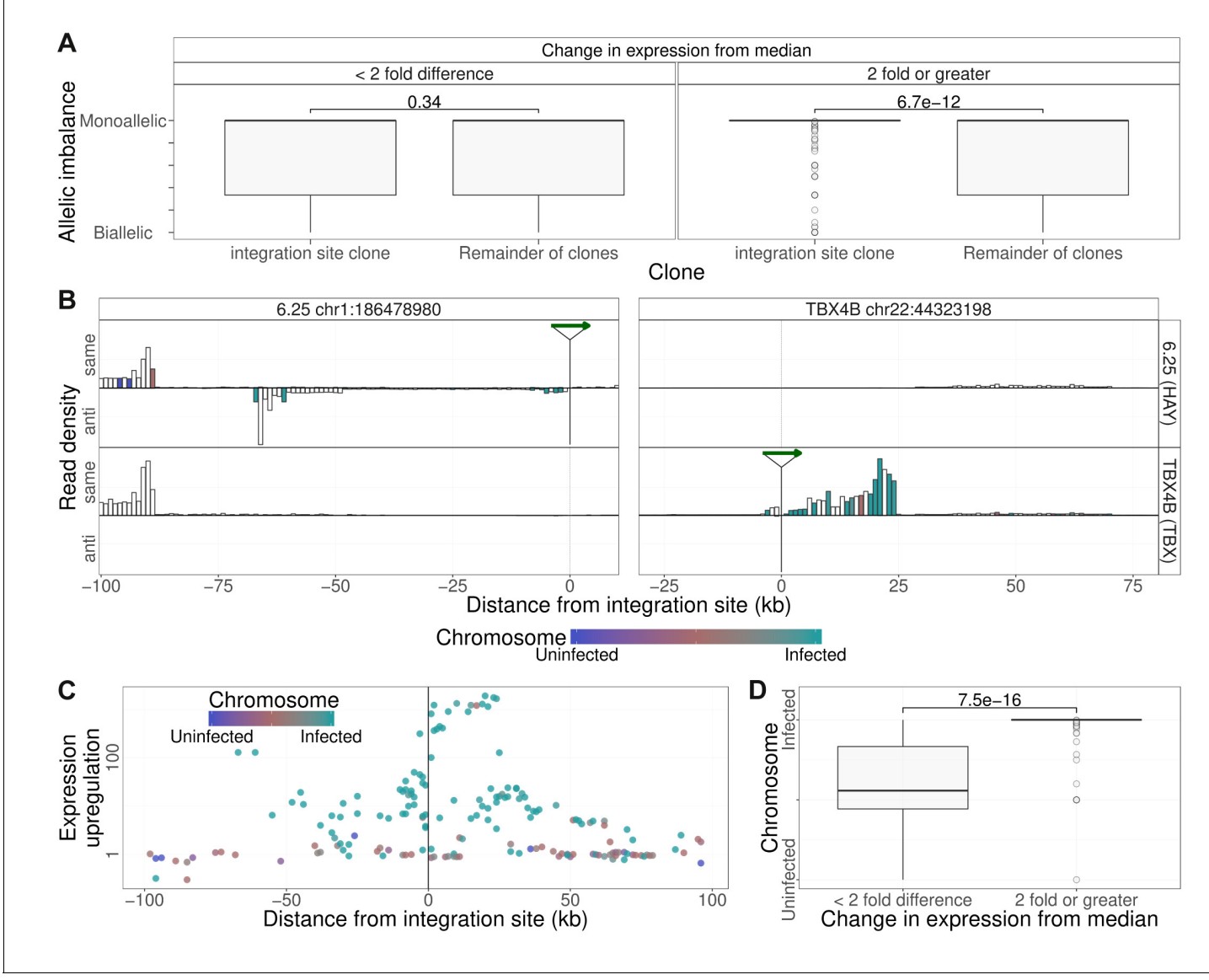

**Figure 5.** Clone-specific host transcription is derived from the infected chromosome. (**A**) Allelic imbalance (AI) denotes the degree of monoallelic usage of identified SNPs: AI = 0 indicates biallelic transcription; AI = 0.5 indicates monoallelic transcription. In each clone, the AI was quantified in transcripts within 2 Mb of the proviral integration sites and compared with the value at that site in all other clones. Clone-specific transcription (transcription density in the clone carrying the provirus, 2-fold or greater than the median; 1 kb bins) was monoallelic; shared transcription was biallelic. While there was no significant difference between the allelic imbalance in those bins for which there was little or no change in transcription from median, for those bins where clone specific expression was observed (two fold or greater increase), the allelic imbalance was significantly greater (more monoallelic) in the integration site clone compared to remainder of clones (p=6.7 * 10$^{-12}$, Wilcoxon test). (**B**) Transcription density depicted as in *Figure 4A*, analysed by haplotype (see *Figure 2*). Columns are coloured by the mean frequency of infected or uninfected alleles (1 kb bins). White columns did not include SNPs that could be assigned to a single haplotype. (**C**) Median ratio of transcription density (log scale) in 1 kb bins containing a heterozygous SNP coloured by the frequency of alleles derived from the infected (green) or uninfected (blue) haplotypes. (**D**) The SNP alleles expressed at $\geq 2 \times$ median level were over-represented in the infected haplotype.

DOI: https://doi.org/10.7554/eLife.36245.026

In addition to the abnormal transcription frequently detected in regions of the host genome contiguous with the provirus, each clone also had a unique pattern of abnormal transcription at non-contiguous, distant sites. The great majority of these clone-specific transcripts came from the chromosome carrying the provirus, not its homologous 'uninfected' chromosome, as shown by the allelic bias observed in the sequence reads (*Figure 5D*). Similarly, the observed allelic bias in the

q4C reads extended to at least 1 Mb upstream and downstream of the provirus (*Figure 2B*). These results are consistent with the notion that the abnormal transcription observed at sites distant from the provirus results from contacts made between the provirus and the host genome.

While certain CTCF-dependent chromatin loops mediate enhancer-promoter interactions, there is growing evidence that the main function of CTCF-dependent loops is structural and is largely shared between cell types, whereas loops formed by other chromatin-binding proteins such as YY1 (*Weintraub et al., 2017*) or chromatin-modifying proteins such as PRC1 (*Eagen et al., 2017*) mediate dynamic enhancer-promoter interactions and thus play a central part in determining cell-type-specific transcription. It remains to be tested whether each respective instance of HTLV-1-associated abnormal host transcription observed here depends on chromatin looping induced by CTCF or another chromatin-binding protein.

Retroviruses have long been known to disrupt host gene expression by insertional mutagenesis, either by integration of the provirus within a gene or by activating expression of a flanking host gene (*Uren et al., 2005*; *Coffin et al., 1997*; *Nusse and Varmus, 1982*); indeed, the first report of insertional oncogenesis was activation of *c-myc* by avian leukosis virus (*Hayward et al., 1981*). Upstream host oncogenes have been shown to be activated by integrated retroviruses, for example mouse mammary tumour virus (*Nusse and Varmus, 1982*) and murine leukemia virus (MLV); activation may occur either by an effect of the proviral enhancer or by readthrough transcription from the LTR into the host genome (*Rasmussen et al., 2010*). Integration downstream of a host oncogene conferred a strong selective advantage on certain clones transduced with a gene therapy vector derived from MLV, resulting in a number of cases of leukemia (*Hacein-Bey-Abina et al., 2008*; *2003*). In mice, it has also been shown that MLV can cause abnormal host transcription by activating a distant host gene (*Lazo et al., 1990*; *Sokol et al., 2014*; *Babaei et al., 2015*). In a study of MLV-induced leukemia, Sokol et al. (*Sokol et al., 2014*) suggested that the MLV proviral enhancer was brought near the oncogene by chromatin looping; if so, it is likely that chance integration of MLV at this particular locus enabled the virus to exploit a normal chromatin loop normally present in the mouse genome. In contrast to this adventitious effect of MLV, our results show that the HTLV-1 provirus itself can cause both abnormal chromatin contacts in cis with distant host loci and abnormal host transcription at distant loci. These findings demonstrate that HTLV-1 has a range of insertional mutagenesis that can extend to the megabase scale.

The altered chromosomal looping and consequent effects on host genome transcription depend on the proviral integration site and the presence of nearby CTCF-BS, and so differ in each infected T-cell clone. The changes in chromatin structure caused by HTLV-1 might not only create novel loops but also destroy pre-existing CTCF-dependent loops that are present in uninfected cells. Further work is needed to investigate the impact of altered looping and transcription on the persistence, infectivity and oncogenic potential of the infected T-cell clones.

We postulate that CTCF confers a survival or replication advantage to HTLV-1, because of the high degree of sequence conservation of the proviral CTCF binding site and the coincidence of the binding site with a border of epigenetic modifications associated with transcriptional activity (*Satou et al., 2016*). However, the nature and mechanism of this putative advantage are not yet known.

Schmidt et al (*Schmidt et al., 2012*) showed that retrotransposons propagated CTCF-BS throughout the genome of several mammalian lineages during evolution; however, present-day exogenous retroviruses have not previously been shown to alter the higher-order structure of host chromatin, to our knowledge. It was recently reported (*Goodman et al., 2018*) that the spumaretrovirus foamy virus encodes a CTCF-BS in its long terminal repeats. The impact of these binding sites on host chromatin structure and transcription have not been investigated; Goodman et al (*Goodman et al., 2018*) suggested that the presence of CTCF bound to the LTRs might block enhancer effects of the foamy virus LTR, and so might account for the low observed degree of genotoxicity of the foamy virus.

The HTLV-1 transcriptional transactivator protein Tax drives expression not only of the proviral plus stand but also of several host genes, notably *CD25* (*IL2RA*), *IFNG*, *IL6*, *IL15*, *GM-CSF*, *TNFB*, and *CCL22* (*Matsuoka and Jeang, 2011b*). While these effects may contribute to the persistence and replication of the virus, and to the risk of the inflammatory and malignant diseases associated with HTLV-1, the effects are not clone-specific. The present results show that, in addition to these generic effects, HTLV-1 has the potential to disrupt host gene expression in cis at both contiguous

and non-contiguous sites. Since the virus infects between $10^4$ and $10^5$ different clones of T cells in a typical host, each carrying the provirus at a different genomic location, we conclude that the virus deregulates tens of thousands of host genes in a typical infected host.

## Materials and methods

### Cell culture, preparation

The HTLV-1-infected T-lymphocyte clones were derived by limiting dilution from peripheral blood mononuclear cells (PBMCs) of donors attending the National Centre for Human Retrovirology (NCHR) at Imperial College Healthcare NHS Trust, St Mary's Hospital, London. All donors gave written informed consent in accordance with the Declaration of Helsinki to donate blood samples to the Communicable Diseases Research Tissue Bank, approved by the UK National Research Ethics Service (15/SC/0089). The derivation of these clones and the genomic insertion site of the single-copy HTLV-1 provirus in each clone were previously reported (*Cook et al., 2012*). The cells were cultured in RPMI-1640 medium (Sigma-Aldrich) with added L-glutamine (Invitrogen), penicillin and streptomycin (Invitrogen), 10% AB human serum (Biowest) at 37°C, 5% $CO_2$. IL-2 (Promokine) was added to the culture every 3 days, and the retroviral integrase inhibitor raltegravir (Selleck) was maintained at 10 µM throughout cell culture, to prevent secondary infection. In addition, the cells were activated every 14 days by the addition of beads coated with antibodies against CD2, CD3 and CD28 (Miltenyi-Biotech). All experiments were carried out on cells harvested on Day 9 of this cycle, after addition of fresh media on Day 8.

### Quantitative circular chromosome conformation capture (q4C); CRISPR/Cas9 modification

We used Cas9 ribonucleoprotein transfection (*Schumann et al., 2015*) to selectively mutate six nucleotides in the core CTCF binding-site in the HTLV-1 provirus and abrogate CTCF binding (*Satou et al., 2016*). q4C–seq libraries were prepared according the 4C protocol by (*van de Werken et al., 2012*) and our protocol for linker-mediated PCR (LM-PCR) (*Gillet et al., 2011*) with a modification (*Supplementary file 2*). Eight million cells were crosslinked in phosphate-buffered saline (PBS) containing 1% formaldehyde for 10 min at room temperature. Cells were lysed using a buffer containing 10 mM Tris-HCl (pH 8.0), 10 mM NaCl, 0.2% NP-40 and complete EDTA-free Protease Inhibitor Cocktail (Roche). The DNA was digested with NlaIII (New England Biolabs, NEB) in the presence of 0.2% SDS, followed by ligation using T4 DNA ligase (NEB) under dilute conditions. The ligated DNA (3 µg) was sheared by sonication with a Covaris S2 instrument; adapters were added and the fragments were subjected to LM-PCR for preparation of 4C–seq libraries. DNA fragments were end-repaired using T4 DNA polymerase, DNA polymerase I Klenow fragment, and T4 polynucleotide kinase (NEB). An adenosine residue was added at the 3′ end of the DNA fragments using Klenow fragment 3′ to 5′ exo– (NEB). A partially double-stranded DNA linker with a specific 6 bp tag was ligated to the DNA ends using a Quick ligation kit (NEB). The linker-ligated product (200 ng per reaction tube) was amplified by a first PCR (PCR1) using the primers HY3 (5′-CTCCTCCTTGTCCTTTAACTCTTCCTC-3′) and Bio4 (5′-TCATGATCAATGGGACGATCA −3′) and Phusion DNA polymerase (NEB). Eight individual PCR1 reactions of 50 µL were prepared for each sample and purified individually using QIAquick PCR Purification Kit (Qiagen) columns. To perform PCR2, 2 µl of the purified PCR1 product was amplified in a 50µl reaction volume between the primers HY12

(5′-AATGATACGGCGACCACCGAGATCTACACCCTCCAAGGATAATAGCCCGTC-3′) and P7 (Illumina). The 8 PCR2 products were combined and purified by QIAquick PCR Purification Kit (Qiagen). The libraries were quantified by qPCR using Illumina primers P5 and P7. Stock 4C libraries were diluted accordingly and clustered on the sequencing flow cell.

The steps in q4C analysis are summarized in *Supplementary file 3*. q4C libraries were sequenced with paired ends (read1 and read 2), with a read length of 100 bp with eight initial dark cycles plus a 6 bp tag read (read 3), on a HiSeq 2500 in Rapid-run mode (Illumina). The sequencing primer was situated in the viewpoint (NlaIII fragment), terminating four bases upstream of the 3′ NlaIII restriction site. These four bases as well as the NlaIII restriction site were used as a filter to ensure sequence specificity, and were subsequently trimmed from the 5′ end of the sequence in the first read of each pair using cutadapt (*Martin, 2011*). Adapter and low quality bases were trimmed using trim galore

(http://www.bioinformatics.babraham.ac.uk/projects/trim_galore/). In order to avoid misalignments due to composite reads (containing NlaIII fragments from multiple contacts), in silico digestion at NlaIII binding sites (CATG) was performed on the reads. 'Digested' Fastq files were subsequently aligned (as single reads) against a merged reference of the human genome (hg19 assembly) and the HTLV-1 genome (accession number AB513134). For each read, only the first digested NlaIII fragment was used for subsequent analysis unless the first fragment was the one directly following the view-point (incomplete digestion) in which case the second fragment was used. Successfully aligned pairs were filtered to remove barcode errors, and only pairs where both read 1 and read two mapped to the same chromosome in convergent orientation were kept. Unique read1-read2 pairs were considered a ligation event: the shear site serves as a unique molecular identifier (Gillet et al., 2011); that is, it identifies a ligation event in a single cell. The number of unique ligation events was then quantified at each ligation site (virus-host genome contact site). At least two biological replicates were analysed from each clone. Where more than two replicates were analysed, the two with the highest library diversity (the highest total number of ligation events) were used in subsequent analyses.

To call chromatin contact sites from the q4C data, we first counted the unique ligation events (comprising distinct ligation and shear positions) in 5 kb overlapping windows (1 kb steps) across the alignments for every clone from each sample. Next we trained a three-state hidden Markov model on all windows with more than one ligation event, for chromosomes containing integration sites, ordered by genomic position, using the Expectation-Maximization (EM) algorithm from the dep-mixS4 package (Visser and Speekenbrink, 2010) to find the initial parameters. We then applied the Viterbi algorithm using this trained model to all individual clones and samples separately.

To define continuous genomic regions of interactions for a given chromosome, we applied a cubic smoothing spline to each sample's respective state space, interpolating over n points across the chromosome with n being 0.25 * the number of overlapping windows in that chromosome. The edges for each peak were defined by the change in sign of the difference of the curve between consecutive points. Peaks were called for a single replicate only if included windows with a minimum of state 2 (probable peak) in both replicates. The peaks were intersected between the replicates using the GenomicRanges package (Lawrence et al., 2013) and intersects which did not include the local maximum of each peak in both replicates were discarded. Peaks which were unique to one clone, were under 50 kb in width and did not overlap the integration site were used in subsequent analysis.

## RNA-seq

RNA-seq libraries were prepared from total RNA of the T cell clones using the TruSeq Stranded mRNA HT Sample Prep Kit according to the manufacturer's instructions, and sequenced with the HiSeq4000 (150 bp paired-end reads). RNA sequencing for each clone was performed using the Illumina platform. Where more than one replicate was sequenced, the one with a larger number of reads was used in subsequent analysis. Data quality was assessed using FastQC (https://www.bioinformatics.babraham.ac.uk/projects/fastqc/) and aligned to the same combined reference (human genome + proviral genome) as described above for q4C analysis, using GSNAP v2017-05-08 (Wu et al., 2010). Read depth analysis was carried out using bedtools v2.25.0 (RRID:SCR_006646) (Quinlan and Hall, 2010) against a series of non-overlapping 1 kb windows up to to 1.5 Mb either side of the respective integration site. The read count for each window was normalized to the total number of aligned reads in the same sample.

## ChIP sequencing (ChIP-seq)

Cells ($1.5 \times 10^7$) were cross-linked with 1% formaldehyde at room temperature for 5 min. Nuclear cell lysates were sonicated with a Covaris S2 and immunoprecipitated using anti-CTCF (Millipore #07–729; RRID:AB_441965) antibody. The ChIP DNA libraries (ChIP and input DNAs) were prepared using NEBNext Ultra II DNA Library Prep Kit for Illumina and Multiplex Oligos for Illumina (New England Biolabs, NEB) according to the manufacturer's instructions. Libraries were sequenced (single-end 50 bp reads) on a HiSeq 2500 (Illumina).

CTCF ChIP libraries from three T-cell clones (two of which carry an HTLV-1 provirus), and a DNA input control were sequenced on the Ilumina platform. Sequence data were trimmed to remove adapters and low-quality bases using TrimGalore, and aligned against the same combined reference

(human + viral genomes) as above, using GSNAP. Duplicates were removed using Picard v.2.9.0 (http://broadinstitute.github.io/picard; RRID:SCR_006525)) and peaks were called against the DNA input control using MACS (*Zhang et al., 2008*), and data from the best replicate (highest number of peaks identified) of each clone were used in downstream analysis. CTCF-BS identified in at least two of the clones examined were used in further analysis. The orientation of CTCF motifs within identified CTCF ChIP peaks was determined using PWMtools PWMscan (Ambrosini G., PWMTools, http://ccg.vital-it.ch/pwmtools) to call the orientation of the highest scoring motif. The orientation of ~79% of CTCF observed binding sites was determined in this way.

## Whole-genome sequencing

Genomic DNA (gDNA) was extracted from each subject's PBMCs and the respective T cell clones using DNeasy Blood and Tissue kit (QIAGEN). Whole-genome sequencing was carried out on PBMC DNA from each subject from whom clones were used in this study, with the exception of subject TBX; DNA from this subject's clone TBX4B was sequenced because PBMC DNA was not available. DNA sequencing was performed on the Ilumina X10 platform, one sample per lane. Alignment against the same combined reference (human + viral genomes) was done using BWA-MEM v0.5.9 (RRID:SCR_010910) (*Li and Durbin, 2009*), and duplicates were removed using Picard. Base calibration (against known dbsnp_135.hg19) and SNP calling for all samples was done using Genome Analyser ToolKit (GATK; RRID:SCR_001876) v3.7 (*DePristo et al., 2011*). Pre-processing, variant discovery and calling followed GATK best practices workflow 3.6. The variant list was filtered to select biallelic variants using GATK SelectVariants and reads per allele were counted by GATK ASEReadCounter, using minimum mapping quality 10, minimum base quality two and minimum depth 10. The B-allele frequency (BAF) of each SNP was calculated as the allele count of alternative base/sum of allele counts of reference and alternative bases. SNPs were defined as heterozygous if the B allele frequency (BAF) was between 0.15 and 0.85.

## Haplotype analysis

Read-aware phasing into haplotypes was performed using SHAPIT v2.r837 (*Delaneau et al., 2013*) which extracts phase-informative reads and assembles them into haplotypes using data modelled on the data from the 1000 Genome Project (https://mathgen.stats.ox.ac.uk/impute/1000GP_Phase3.html). RNASeq data were aligned to the combined reference using GSNAP (a variant-aware aligner, known to reduce reference bias [*DePristo et al., 2011*]); duplicates were removed using Picard and the alleles counted for each biallelic variant using GATK ASEReadCounter, using a minimum mapping quality of 10 and minimum base quality of 2. Allelic imbalance (AI) in q4C and RNA-seq data was calculated by the formula $AI = abs(0.5 - BAF)$; AI ranges between 0 (biallelic) and 0.5 (monoallelic expression).

## Statistics

Nonparametric tests were used to examine the association between the number of CTCF-BS and the q4C peak height (Wilcoxon) and the number of q4C peaks (Spearman). The difference in the frequency of q4C peaks at a given distance from the provirus was tested using a chi-squared goodness of fit test. The correlation between the number of q4C peaks and the density of CTCF-BS was examined using Pearson's test. The curves showing the relationship between allelic imbalance and genomic distance (*Figure 2B*) were computed using LOESS regression.

## Data availability

Sequence data have been deposited at the European Genome-phenome Archive (EGA, http://www.ebi.ac.uk/ega/), hosted by the European Bioinformatics Institute (accession number EGA S00001002259). Custom scripts used to analyse this data are available at https://github.com/ImperialCollegeLondon/q4C-2018 (*Melamed, 2018*; copy archived at https://github.com/elifesciences-publications/q4C-2018).

## Acknowledgements

We thank Graham Taylor, Lucy Cook, the donors and research nurses in the National Centre for Human Retrovirology, Imperial College. We are grateful to Yorifumi Satou, Heather Niederer, Aileen Rowan, Jocelyn Turpin and other members of the Bangham laboratory for helpful discussion. DNA sequencing (q4C and ChIP assays) was performed in the Genomics Facility in the MRC London Institute for Medical Sciences, Hammersmith Hospital, London, UK (Laurence Game, Marian Dore). RNA sequencing was performed in the Institute of Child Health, University College London (Mike Hubank, Kerra Pearce, Tony Brooks and Masahiro Ono) and in the Wellcome Trust Centre for Human Genetics, Oxford, UK. Whole-genome sequencing was performed in the Wellcome Trust Sanger Institute, Hinxton, Cambridge, UK. Data analysis was performed using both Imperial High-Performance Computing resources and the computing cluster at the European Bioinformatics Institute, Hinxton, Cambridge, UK. This work was supported by the Wellcome Trust (https://wellcome.ac.uk/) (CRMB Senior Investigator Award WT100291MA), the Medical Research Council (MRC, http://www.mrc.ac.uk/) (MR/K019090/1), the Imperial National Institute for Health Research Biomedical Research Centre (http://imperialbrc.org/), and the Naito Foundation, Japan (https://www.naito-f.or.jp/en/).

## Additional information

### Funding

| Funder | Grant reference number | Author |
| --- | --- | --- |
| Wellcome | WT100291MA | Anat Melamed<br>Hiroko Yaguchi<br>Michi Miura<br>Aviva Witkover<br>Charles RM Bangham |
| Naito Foundation | | Michi Miura |
| Medical Research Council | MR/K019090/1 | Charles RM Bangham |
| National Institute for Health Research Imperial Biomedical Research Centre | | Charles RM Bangham |

The funders had no role in study design, data collection and interpretation, or the decision to submit the work for publication.

### Author contributions

Anat Melamed, Conceptualization, Data curation, Software, Formal analysis, Supervision, Funding acquisition, Validation, Investigation, Visualization, Methodology, Writing—original draft, Project administration, Writing—review and editing; Hiroko Yaguchi, Data curation, Conceptualization, Supervision, Formal analysis, Validation, Investigation, Visualization, Methodology, Writing—original draft, Writing—review and editing; Michi Miura, Resources, Formal analysis, Validation, Investigation, Methodology, Writing—original draft, Writing—review and editing; Aviva Witkover, Resources, Validation, Investigation, Methodology, Writing—original draft, Writing—review and editing; Tomas W Fitzgerald, Software, Formal analysis, Validation, Investigation, Methodology, Writing—original draft, Writing—review and editing; Ewan Birney, Conceptualization, Resources, Supervision, Methodology, Project administration, Writing—review and editing; Charles RM Bangham, Conceptualization, Resources, Supervision, Funding acquisition, Methodology, Writing—original draft, Project administration, Writing—review and editing

### Author ORCIDs

Anat Melamed (iD) https://orcid.org/0000-0002-7312-3138
Hiroko Yaguchi (iD) https://orcid.org/0000-0002-6184-0033
Michi Miura (iD) http://orcid.org/0000-0002-6943-3782
Tomas W Fitzgerald (iD) http://orcid.org/0000-0002-2370-8496

Ewan Birney (iD) http://orcid.org/0000-0001-8314-8497
Charles RM Bangham (iD) http://orcid.org/0000-0003-2624-3599

## Ethics

Human subjects: All donors gave written informed consent in accordance with the Declaration of Helsinki to donate blood samples to the Communicable Diseases Research Tissue Bank, approved by the UK National Research Ethics Service (15/SC/0089).

## Decision letter and Author response

Decision letter https://doi.org/10.7554/eLife.36245.034
Author response https://doi.org/10.7554/eLife.36245.035

## Additional files

### Supplementary files

• Supplementary file 1. T cell clones used. Extended data on clones shown in *Table 1*. All subjects are HTLV-1 carriers with HAM/TSP, except for HAY who is an asymptomatic carrier of HTLV-1. *tax* expression of 'high' or 'low' denotes whether the frequency of plus-strand viral transcripts was higher or lower than the median, respectively.
DOI: https://doi.org/10.7554/eLife.36245.027

• Supplementary file 2. Schematic diagram to compare conventional 4C (A) and q4C (B) protocol. In the conventional protocol (A), after digesting the crosslinked chromatin with the first restriction enzyme (1 st RE) and ligating the free ends, the DNA was digested with a second restriction enzyme (second RE) followed by religation and inverse PCR to amplify viewpoint (VP)-linked genomic regions. In q4C, we modified the 4C protocol (*Krijger and de Laat, 2016*) by applying the approach used in our previously described linker-mediated (LM)-PCR protocol (*Gillet et al., 2011*) for identifying and quantifying proviral integration sites. In q4C, instead of the secondary restriction enzyme, sonication is used to process DNA circles. Linkers with a 6 bp specific tag was added to sonicated DNA. The end of the VP and a fragment of genomic DNA were amplified by LM-PCR. In this example, three ligation events occurred between the VP (red) and a genomic region (green) at the ligation site I and one event at the ligation site II (yellow). Because the DNA shear site is (approximately) random, the amplicon from each cell has a different shear site. The abundance of ligation events at each respective ligation site is quantified by counting the number of different shear sites.
DOI: https://doi.org/10.7554/eLife.36245.028

• Supplementary file 3. q4C data analysis steps. Summary of main steps in the analysis steps of q4C data. See Materials and methods for details.
DOI: https://doi.org/10.7554/eLife.36245.029

• Transparent reporting form
DOI: https://doi.org/10.7554/eLife.36245.030

### Data availability

Sequence data have been deposited at the European Genome-phenome Archive (EGA, http://www.ebi.ac.uk/ega/), hosted by the European Bioinformatics Institute (accession number EGA S00001002259). Custom scripts used to analyse this data are available at https://github.com/ImperialCollegeLondon/q4C-2018.

The following dataset was generated:

| Author(s) | Year | Dataset title | Dataset URL | Database, license, and accessibility information |
|---|---|---|---|---|
| Melamed A, Yaguchi H, Miura M, Witover A, Fitzgerald TW, Birney E, Bangham CRM | 2018 | The human leukemia virus HTLV-1 alters the structure and transcription of host chromatin in cis | https://www.ebi.ac.uk/ega/studies/EGA S00001002259 | Publicly available at the European Genome-phenome Archive (accession no: EGAS00001002259) |

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
