## [Decision Letter]

Thank you for submitting your article "The human leukemia virus HTLV-1 alters the structure and transcription of host chromatin *in cis*" for consideration by *eLife*. Your article has been reviewed by two peer reviewers, and the evaluation has been overseen by a Reviewing Editor and Wenhui Li as the Senior Editor. The reviewers have opted to remain anonymous.

The reviewers have discussed the reviews with one another and the Reviewing Editor has drafted this decision to help you prepare a revised submission.

I apologize for the delay, but I am pleased to tell you that I have now obtained thoughtful reviews, and all of the reviewers were positive about your manuscript. Both reviewers had suggestions that I believe will strengthen the message of your manuscript. I believe that appropriate responses can be provided without any further experimentation. Please respond to each of the reviewers comments in an itemized response letter, along with a revised manuscript.

*Reviewer #1:*

The authors have studied how the presence of a binding site for the chromatin architectural protein CTCF within the HTLV-1 provirus affects chromatin loops within the host cell nucleus. Using an elegant adaptation of the circular chromosome conformation capture (4C) technique, they identify chromosomal regions that form loops with the CTCF binding region of the provirus. The loops are shown to form in cis and not in trans and to alter transcription of host genes near the integration site and distant sites within a loop.

Overall, this is an interesting and well performed study. The manuscript could be improved by considering the following points:

1) The results clearly show that the CTCF binding site in the viral genome can participate in looping with cis host genomic elements, but whether this alters host genomic looping as the authors suggest is less clear. Can the authors define the loop patterns that would be present in the region of the integration site when no provirus is present?

2) It would be helpful to provide a schematic of the authors' modifications to the 4C method.

3) How have the quantitative aspects of this method been validated?

4) The altered chromosomal looping and consequent effects on host gene transcription will be different in each clone with a different integration site. Overall, it is not clear how this affects HTLV-1 persistence or oncogenic potential. Some discussion of these points would strengthen the manuscript.

*Reviewer #2:*

It has been known for many years that insertional activation (or, rarely, inactivation) of gene expression is the principal mechanism by which they cause cancer, a fact which enabled scientists to identify and study in simple animal models, genes which turned out to be of central importance to human cancer. HTLV has always been a somewhat embarrassing exception; in only a few cases has oncogenic disruption of genes near the site of integration. The elegant study by Melamed et al., submitted to *eLife*, promises to provide an answer to this quandary, and to significantly advance our understanding of important aspects of the HTLV-host interaction.

Using a panel of 10 clonal HTLV-infected cell lines, from infected patients, each with a single provirus, the authors developed and used a novel DNEA looping assay to show that proviral sequences, which contain a CTCF binding site can be brought into juxtaposition with regions of neighboring chromosomal DNA up to as many as 4 Mbp away, and that this looping, by bringing viral enhancer elements near cellular promoters, can affect the transcription of genes on the same chromosome, but at a considerable distance, providing a likely explanation of why, in most cases, classical oncogene activation patterns of adjacent genes are not seen in HTLV-induced lymphoma, even if the underlying mechanism might be almost the same.

All in all, I found this to be a well-written and convincing study, certainly worthy of publication in *eLife*. My comments are mostly just points that might benefit from further discussion.

1) Although the primary observations seem quite compelling, the authors seem to tiptoe around the role of chromosomal looping in oncogene activation. There must be additional evidence in support of such a model that they can cite, such as the existence of oncogenic cell lines with aberrant expression of known oncogenes within a few Mbp of an HTLV-1 provirus. The focus of this study on non oncogenic lines makes it seem like they are splitting off an important part of the story.

2) It seems highly unlikely that the CTCF binding site is in the provirus by accident. Some discussion – even speculation – to this point would be very welcome.

---

## [Author Response]

Reviewer #1:[…] Overall, this is an interesting and well performed study. The manuscript could be improved by considering the following points:1) The results clearly show that the CTCF binding site in the viral genome can participate in looping with cis host genomic elements, but whether this alters host genomic looping as the authors suggest is less clear. Can the authors define the loop patterns that would be present in the region of the integration site when no provirus is present?

This is indeed an interesting point: it is possible that HTLV-1 destroys normal chromatin loops as well as creating abnormal loops. However, the appropriate method to use to systematically map the loop patterns in the normal genome is HiC, which in principle identifies loops (albeit with limited sensitivity) between any two genomic regions (‘all to all’ mapping), whereas 4C identifies loops made between a defined ‘viewpoint’ and all other regions (‘one to all’ mapping). It is outside the scope of this study to carry out HiC analysis in addition to the 4C.

2) It would be helpful to provide a schematic of the authors' modifications to the 4C method.

Thank you for the suggestion. We have made a schematic diagram to compare conventional 4C and our modified version, q4C, and this is included as Supplementary file 2.

3) How have the quantitative aspects of this method been validated?

Quantification of chromatin looping. 4C (and indeed other ‘C techniques’) gives relative quantification of chromatin looping: in the absence of single-cell 4C – which no-one has yet developed – absolute quantification of loop frequency is not yet possible. Therefore, validation of the (relative) loop frequency depends primarily on the reproducibility of peak identification in independent biological replicate experiments, as we report in the present Results. The improvement in quantification introduced by the use of sonication instead of restriction enzyme digestion was demonstrated in our high-throughput analysis of retroviral integration sites (Blood 2011: 117, 3113; Bioinformatics 2012: 28, 755; et seq.)

4) The altered chromosomal looping and consequent effects on host gene transcription will be different in each clone with a different integration site. Overall, it is not clear how this affects HTLV-1 persistence or oncogenic potential. Some discussion of these points would strengthen the manuscript.

The impact of the provirus on chromatin looping and host transcription is indeed clone-specific. Because of the great clonal diversity of HTLV-1-infected cells in each host, identifying the impact of altered chromatin looping on HTLV-1 persistence or oncogenic potential will depend largely on bioinformatic analysis: in particular, to compare the association between CTCF binding sites and proviral integration sites between clones infected in vitro – in the absence of in vivo selection – and those that persist in vivo, and between malignant infected (ATL) clones and non-malignant clones. We are now embarking on this analysis. To emphasize the point raised by the reviewer, we have added the following sentence to the Discussion section:

“The altered chromosomal looping and consequent effects on host genome transcription depend on the proviral integration site, and so differ in each infected T-cell clone. Further work is needed to investigate the impact of altered looping and transcription on the persistence, infectivity and oncogenic potential of the infected T-cell clones.”

Reviewer #2:[…] All in all, I found this to be a well-written and convincing study, certainly worthy of publication in eLife. My comments are mostly just points that might benefit from further discussion.1) Although the primary observations seem quite compelling, the authors seem to tiptoe around the role of chromosomal looping in oncogene activation. There must be additional evidence in support of such a model that they can cite, such as the existence of oncogenic cell lines with aberrant expression of known oncogenes within a few Mbp of an HTLV-1 provirus. The focus of this study on non oncogenic lines makes it seem like they are splitting off an important part of the story.

Relationship of proviral integration to oncogene activation. This is of course a very important point. However, the chief aim of the present paper was to show that insertional mutagenesis – both local and at distant sites – is a general feature of HTLV-1 infection, and yet does not usually cause oncogenic transformation. We now plan to apply the techniques reported here to analyse chromatin looping and host transcription in cases of adult T-cell leukemia/lymphoma. In this planned study, we will focus on primary cells from ATL cases, rather than long-term in vitro cell lines, because ATL clones are notoriously difficult to propagate in vitro, so the relationship between such in vitro cell lines and primary ATL clones is at best uncertain. See also the response to reviewer 1, point 4.

2) It seems highly unlikely that the CTCF binding site is in the provirus by accident. Some discussion – even speculation – to this point would be very welcome.

We agree wholeheartedly with the reviewer’s implication that CTCF binding must confer some benefit on HTLV-1. However, this has proved a very difficult problem. To clarify the point, as the reviewer suggests, we have added the following short paragraph to the Discussion:

“We postulate that CTCF confers a survival or replication advantage to HTLV-1, because of the high degree of sequence conservation of the proviral CTCF binding site and the coincidence of the binding site with a border of epigenetic modifications associated with transcriptional activity [Satou et al., 2016]. However, the nature and mechanism of this putative advantage are not yet known.”